

# Morphometric relationships and seasonal variation in size, weight, and a condition index of post-settlement stages of the Caribbean spiny lobster

Rogelio Martínez-Calderón[1,2], Enrique Lozano-Álvarez[1] and Patricia Briones-Fourzán[1]

[1] Instituto de Ciencias del Mar y Limnología, Unidad Académica de Sistemas Arrecifales, Universidad Nacional Autónoma de México, Puerto Morelos, Quintana Roo, Mexico
[2] Posgrado en Ciencias del Mar y Limnología, Universidad Nacional Autónoma de México, Ciudad de México, Mexico

Corresponding author
Patricia Briones-Fourzán,
briones@cmarl.unam.mx

## ABSTRACT

Spiny lobsters have a protracted pelagic, oceanic larval phase. The final larval stage metamorphoses into a non-feeding postlarva (puerulus) that actively swims towards the coast to settle in shallow habitats and does not resume feeding until after the molt into the first-stage juvenile. Therefore, the body dimensions and nutritional condition of both settled pueruli and first juveniles are likely to vary over time, potentially playing a crucial role in the recruitment to the benthic population. We compared carapace length (CL), height (CH), and width (CW); total length (TL), and body weight (W) between pueruli and first juveniles of the Caribbean spiny lobster, *Panulirus argus*, as well as morphometric relationships between both developmental stages. Except for CL, all other dimensions were larger in first juveniles, but more markedly CH and W. The slopes of the CH vs CL, CW vs CL, and W vs CL regressions differed significantly between stages, and all log-transformed relationships showed isometry in both stages, except for the CH vs CL relationship, which showed positive allometry. These results reflect a morphological change from the flatter, more streamlined body of the puerulus, to the heavier, more cylindrical body of the juvenile. We also analyzed seasonal variations in CL, W, the W/CL index (a morphometric condition index), and a modified W/CL index (i.e. after controlling for a significant effect of CL) of both stages using individuals monthly collected over 12 consecutive seasons (Autumn 2010–Summer 2013). In both stages, all three variables exhibited significant seasonal variation. For pueruli, the modified W/CL index differed from average in only two seasons, winter 2011 (higher) and summer 2013 (lower), but showed great within-season variation (larger coefficients of variation, CV), potentially reflecting variability in nutritional condition of larvae prior to metamorphosis and in the distances swum by individual pueruli to the settlement habitats. For first juveniles, the modified W/CL index was higher than average in winter and spring 2011, and lower in autumn 2011 and winter 2012, but showed less within season variation (smaller CVs), suggesting a combination of carry-over effects of puerulus condition and effects of local conditions (e.g., food availability and predation risk). These findings warrant further investigation into factors potentially decoupling settlement from recruitment processes.

## INTRODUCTION

Spiny lobsters (family Palinuridae) are large decapod crustaceans that constitute valuable fishing resources wherever they occur (*Holthuis, 1991*). Palinurids have a protracted oceanic larval phase that consists of multiple flattened, leaf-like larval stages (phyllosoma) morphologically very different from the adults (*Phillips et al., 2006*). The final phyllosoma stage metamorphoses into a highly transparent swimming postlarva (the puerulus), morphologically similar to the adult. This metamorphosis has been considered the most dramatic morphological transformation in a single molt across crustaceans (*Gurney, 1942*). The puerulus is a secondary lecithotrophic (i.e., non-feeding) phase, consisting of a single stage that actively swims to the coast, where it settles in shallow habitats (*Phillips et al., 2006*). After settlement, the transparent postlarvae gradually become pigmented and the molt from puerulus to first-stage juvenile (hereafter ''first juvenile'') occurs within days or weeks. The first juveniles resume feeding shortly after molting.

During the post-settlement molt from puerulus to the first juvenile stage, the body of the lobster undergoes substantial changes that make it more suitable for a benthic existence. Most papers reporting on such changes have put emphasis on the development of the mouthparts and other external and internal structures (e.g., *Lemmens & Knott, 1994*; *Nishida, Takahashi & Kittaka, 1995*; *Abrunhosa & Kittaka, 1997*; *Cox, Jeffs & Davis, 2008*), but changes in morphometric dimensions and allometry have been much less explored (*Grobler & Ndjaula, 2001*; *Groeneveld et al., 2010*; *Anderson et al., 2013*).

The Caribbean spiny lobster, *Panulirus argus* (Latreille, 1804) occurs throughout the wider Caribbean region and is by far the most heavily fished of the 24 extant species of spiny lobsters (*Phillips et al., 2013*). The larval phase of *P. argus* comprises 10 phyllosoma stages (*Goldstein et al., 2008*) and metamorphosis occurs in oceanic waters. The puerulus swims to the coast and settles in shallow vegetated substrates (macroalgal beds, seagrass meadows, coastal mangroves), where the early benthic juveniles remain for a few months before shifting to coral reef habitats (*Butler IV & Herrnkind, 2000*). *Lewis, Moore & Babis (1952)* described the puerulus and first few juvenile stages of *P. argus*, and *Anderson et al. (2013)* examined allometric growth of defensive and reproductive structures through ontogeny in *P. argus*, but to our knowledge the relationships between weight and size dimensions and allometry in postlarvae and first juveniles of this species have not been explored.

Artificial collectors mimicking the natural settlement habitats have been devised to monitor the level of puerulus settlement over the long term (*Phillips & Booth, 1994*). In a few spiny lobster species from temperate or cold water regions, in which reproduction and postlarval settlement are strongly seasonal, the settlement data have shown relationships with the commercial catch after a lag of several years, opening the possibility of developing catch predictive models (reviewed in *Phillips et al., 2013*). However, this has not been the case for *P. argus,* suggesting that for tropical species characterized by year-long reproduction
and postlarval settlement, a number of factors potentially mask or decouple the relationship between levels of settlement and abundance of subsequent benthic phases (*Herrnkind & Butler IV, 1994*; *Briones-Fourzán, Baeza-Martínez & Lozano-Álvarez, 2009*). These factors may include quality of settlement habitat, food availability, and predation mortality, as well as variability in nutritional condition of settling pueruli.

In crustaceans, the nutritional condition is the extent to which an individual has accumulated reserves of nutrients to allow normal physiological function and growth (*Moore, Smith & Loneragan, 2000*). Nutritional condition can be an indicator of past foraging success and ability to cope with environmental pressures, and can greatly affect population dynamics (*Jakob, Marshall & Uetz, 1996*). Because pueruli do not feed, their post-settlement nutritional condition will depend on the nutrient reserves accumulated prior to metamorphosis and the energetic cost of swimming to the coastal settlement habitats (*McWilliam & Phillips, 1997*; *García-Echauri & Jeffs, in press*). If pueruli settle with a very low nutritional condition, they may die before, during, or soon after molting into first juveniles, precluding their recruitment to the benthic populations (*Lemmens, 1994*; *Jeffs, Nichols & Bruce, 2001*; *Fitzgibbon, Jeffs & Battaglene, 2014*). After molting, the first juveniles need to resume feeding to replenish their energy sources, but this may depend on local availability of food and predation risk (*Smith & Herrnkind, 1992*; *Weiss, Lozano-Álvarez & Briones-Fourzán, 2008*), which may further reduce the nutritional condition of first juveniles, potentially precluding their molt into second-stage juveniles (*Espinosa-Magaña, Lozano-Álvarez & Briones-Fourzán, 2017*). Therefore, monitoring the levels of nutritional condition of benthic pueruli and first juveniles may provide insight into the potential factors decoupling the relationships between the levels of settlement and the abundance of subsequent benthic phases of *P. argus*.

Several studies have examined the use of nutrients (lipids, proteins, and carbohydrates) during the transition from phyllosoma to puerulus (e.g., *Jeffs, Wilmott & Wells, 1999*; *Limbourn & Nichols, 2009*; *Espinosa-Magaña et al., in press*), or the contents of these nutrients as biochemical indicators of nutritional condition over time in settled pueruli and first juveniles (*Limbourn et al., 2009*). However, this type of biochemical analyses destroys the animals and can be expensive and time-consuming, making their use impractical to monitor condition over the long term. Other biochemical nutritional indices, such as blood refractive index, can be accurate but are strongly affected by the molt cycle (*Oliver & MacDiarmid, 2001*; *Lorenzon, Martins & Ferrero, 2011*). Alternative morphometric techniques avoid injuring the animals and can be more practical to use over the long term (*Jakob, Marshall & Uetz, 1996*). Some studies have found significant variation in the size of benthic (settled) pueruli over time (e.g., *Booth & Tarring, 1986*; *Yeung et al., 2001*; *Groeneveld et al., 2010*), but size alone is not a good reflection of nutritional condition.

One of the simplest morphometric indices of condition is the "ratio index", which has long been used to assess nutritional condition in fishes (*Froese, 2006*) and is calculated as body mass divided by a linear dimension of body size. The ratio index has also been used in crustaceans such as prawns and lobsters, in which this index is usually estimated as the total weight of the animal (W) divided by its carapace length (CL). CL is typically preferred over total length because, unlike the highly flexible abdomen, the rigid carapace can be

measured with less error. Therefore, we herein refer to this index as the "W/CL index". The W/CL index has been shown to reflect nutritional condition in shrimps (*Araneda, Pérez & Gasca-Leyva, 2008*), prawns (*Da Rocha et al., 2015*), and spiny lobsters (*Robertson, Butler & Dobbs, 2000*; *Oliver & MacDiarmid, 2001*; *Briones-Fourzán, Baeza-Martínez & Lozano-Álvarez, 2009*; *Lopeztegui-Castillo, Capetillo-Pinar & Betanzos-Vega, 2012*). This is because, unlike the blood refraction index, the W/CL index is independent of the molt cycle and may be a more direct reflection of nutritional health (*Oliver & MacDiarmid, 2001*).

The W/CL index is also useful to compare the nutritional status between groups or populations (e.g., *Robertson, Butler & Dobbs, 2000*), although *Briones-Fourzán, Baeza-Martínez & Lozano-Álvarez (2009)* showed that its use for these purposes was only warranted when the size range of the groups was similar because the W/CL index increases with size. Given that pueruli and first juveniles have a small and similar size range, the W/CL index could be useful for long-term monitoring of the condition of these early benthic stages of spiny lobsters, but to our knowledge it has not yet been used for those purposes. Therefore, the aims of the present study were twofold: to compare several morphometric relationships between pueruli and first juveniles of *P. argus* and examine allometric changes during this ontogenetic transition, and to examine seasonal variations in size, weight, and the W/CL index of recently settled pueruli and first juveniles of *P. argus*. We predicted that both pueruli and first juveniles would exhibit significant variation in all three variables, the former because of the varying environmental conditions during their long larval phase and the potential variation in the distance swum by different individuals, and the latter because of carry-over effects (*Marshall & Morgan, 2011*) in conjunction with local effects after the post-settlement molt (e.g., predation risk, changes in food availability).

## MATERIALS AND METHODS

### Sampling of organisms

Settled pueruli (transparent and pigmented) and first juveniles of *P. argus* were obtained from GuSi postlarval collectors (*Gutiérrez-Carbonell, Simonín-Díaz & Briones-Fourzán, 1992*) that are used to monitor monthly postlarval settlement indices on the Caribbean coast of Mexico (eastern coast of the Yucatan peninsula) (*Briones-Fourzán, Candela & Lozano-Álvarez, 2008*). These collectors, in sets of six, were deployed leeward of the coral reef tract at two locations: Puerto Morelos (at 20°50.9′N, 86°52.1′W) and Bahía de la Ascensión (at 19°49.8′N, 87°27.1′W). The GuSi collectors simulate intricate marine vegetation and are colonized by very large numbers of small invertebrates (mostly <5 mm long), such as isopods, amphipods, shrimps, crabs, ostracods, harpacticoid copepods, small gastropods, nudibranchs, ophiuroids, and many types of worms (*Mendoza-Barrera & Cabrera, 1998*), thus providing abundant natural prey for the first juveniles after the post-settlement molt (*Marx & Herrnkind, 1985*; *Lalana & Ortiz, 1991*). A permit for collection was issued by Comisión Nacional de Acuacultura y Pesca (DGOPA-06695.190612.1737). We had access to all samples taken from collectors at Puerto Morelos between November 2010 and September 2013, but only to samples taken from collectors at Bahía de la Ascensión between

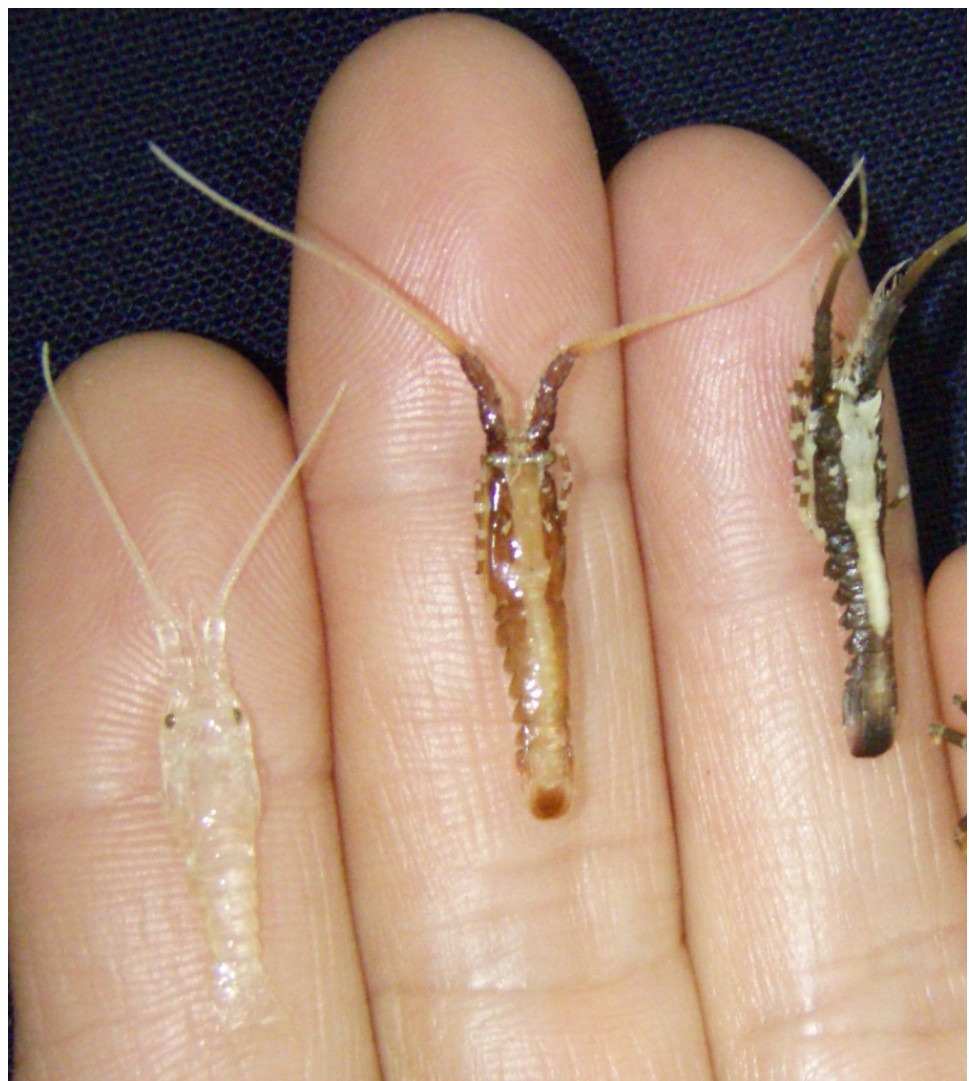

**Figure 1** **Developmental stages of *Panulirus argus* considered in the present study.** From left to right: transparent puerulus, pigmented puerulus, and first-stage juvenile of the Caribbean spiny lobster *Panulirus argus*. The term "pueruli" in the text refers to transparent + pigmented pueruli. (Photo: F Negrete-Soto, with permission).

August 2012 and September 2013. Individuals were transported in aerated seawater to the laboratory within 1 to 3 h of collection, and categorized into transparent pueruli (which are completely transparent except for the eyes), pigmented pueruli (with dark stripes of coloration in the new cuticle, visible below the old, transparent cuticle), first juveniles (Fig. 1), and older juveniles. First juveniles were distinguished from older juveniles based on morphological differences described by *Lewis, Moore & Babis (1952)*, mainly the length of the medial flagella (inner branch) of the antennules relative to the lateral flagella (outer branch), the appearance of the grooves in the abdominal tergites, and the degree of development of the pleopods. Older juveniles were not further considered in this study.

## Morphometric relationships and allometry

Morphometric relationships allow testing the relationship between two variables and predicting the value of one (Y) from the other (X), whereas allometry is useful to examine how one variable scales with another (*Warton et al., 2006*). To examine morphometric relationships and allometry, we used individuals collected from both locations between August 2012 and September 2013. For these individuals, we measured the following length dimensions: carapace length (CL, from between the rostral horns to the posterior edge of the cephalothorax), carapace height (CH, across the mid-point between the bases of the first pair of pereopods and the highest point in the cephalothorax), carapace width (CW, across the widest part of the cephalothorax), and total length (TL, from between the rostral horns to the posterior edge of the telson) with digital Vernier calipers ($\pm$0.01 mm). All length dimensions were measured under a stereoscopic microscope to reduce measurement error. We also measured total wet weight (W) of individuals with a digital scale ($\pm$0.001 g) after blotting excess moist.

Exploratory analyses revealed that transparent and pigmented pueruli did not differ significantly in any of the body dimensions that we measured (Student's $t$-tests, $p > 0.05$ in all cases); therefore, we pooled their data into one category ("pueruli") for further analyses. Prior to regression analyses, we used separate general lineal models (GLM) to test for potential effects of location (Puerto Morelos and Bahía de la Ascensión) and developmental stage (pueruli and first juveniles) on each body dimension. As the effect of location was not significant (see 'Results'), we pooled the data from both locations and used ordinary least-squares regression (OLR) to examine morphometric relationships (*Warton et al., 2006*). For each stage, CH, CW, TL, and W were regressed against CL, which can be measured with less error than TL. For each relationship, the slopes of regressions were compared between stages with Student's $t$-tests; if the slopes did not differ significantly, we compared the elevations (*Zar, 1999*). We then used the log-transformed data of all dimensions to test for allometry, which involves testing if the slope equals a specific value (i.e., isometry: $b = 1$ for length-length relationships and $b = 3$ for length-weight relationships) (*Hartnoll, 1982*). The appropriate method to estimate slopes for this purpose is the reduced major axis regression (RMA, also known as standardized major axis regression) (*Warton et al., 2006*). Slopes were then tested for departures from isometry with Student's $t$-tests.

## Seasonal variation in CL, W, and W/CL index

To examine seasonal variation in size, weight, and the W/CL index, we used exclusively the samples from Puerto Morelos, as this location was sampled uninterruptedly for three years. Given that the monthly catch of the different stages tends to be low or even zero (*Briones-Fourzán, Candela & Lozano-Álvarez, 2008*), we pooled the data by astronomical season (spring: March 20 to June 20; summer: June 21 to September 21; autumn: September 22 to December 20; winter: December 21 to March 19) based on the day of the month when the collectors were sampled. This procedure resulted in data from 12 consecutive seasons.

For these samples, we measured CL and W in the same way as for morphometric analyses. These dimensions allowed for the estimation of the W/CL index. We used separate GLMs to examine the potential effects of season (12 levels) and stage (2 levels, pueruli and first juveniles) on each variable (CL, W, and W/CL). However, because the W/CL index is affected by size (*Briones-Fourzán, Baeza-Martínez & Lozano-Álvarez, 2009*), the seasonal variation in the W/CL index was further evaluated with an analysis of covariance (ANCOVA), using CL as a covariate to control for its significant effect. Least squares means and confidence intervals for seasonal W/CL were then computed for covariates at their means, yielding a modified W/CL index. For each stage and season, the coefficient of variation ($CV = \left(\frac{SD}{mean}\right) \times 100$) was estimated as a measure of within-seasonal variability.

Statistical analyses were done with the software Statistica v.10 (StatSoft, Inc., Tulsa, OK, USA) except for the OLS and RMA regression analyses, which were done with the software PAST v.3.20 (*Hammer, Harper & Ryan, 2001*). In all cases, results were considered significant if $p < 0.05$.

## RESULTS

### Size distribution, morphometric relationships, and allometry

Exploratory analyses revealed that transparent and pigmented pueruli did not differ significantly in any of the body dimensions that we measured (Student's $t$-tests, $p > 0.05$ in all cases); therefore, we pooled their data into one category ("pueruli") for further analyses. Between August 2012 and September 2013, we obtained 606 individuals, 380 pueruli (278 from Puerto Morelos and 102 from Bahía de la Ascensión) and 226 first juveniles (154 from Puerto Morelos and 72 from Bahía de la Ascensión). Size ranges of pueruli and first juveniles were 5.01–6.90 mm CL and 5.31–6.98 mm CL, respectively (Fig. 2). Results from the GLMs examining the effects of location and developmental stage on body dimensions (Table 1) indicated that, except for CL, which did not differ significantly between pueruli and first juveniles, all other body dimensions were significantly affected by stage, but not by location, and the interaction term was also not significant. Therefore, the data from both locations were pooled prior to regression analyses.

In general, morphometric relationships of pueruli exhibited a greater dispersion of data around the OLS regression line (Figs. 3A–3D), resulting in lower coefficients of determination than for first-stage juveniles (Table 2). In the TL vs CL relationship, the slopes of the OLS regressions did not differ significantly between pueruli and first juveniles, but the elevations did (Fig. 3A, Table 2). The slopes of all other regressions differed significantly between both stages (Table 2). In particular, W (Fig. 3B) and CH (Fig. 3C) increased more steeply with increasing CL in juveniles compared to pueruli. Results of RMA regressions revealed that for both stages, all relationships with log-transformed data exhibited isometry, except for the Ln CH vs Ln CL relationship in both stages and the Ln CW vs Ln CL relationship in pueruli, which showed positive allometry (Table 3).
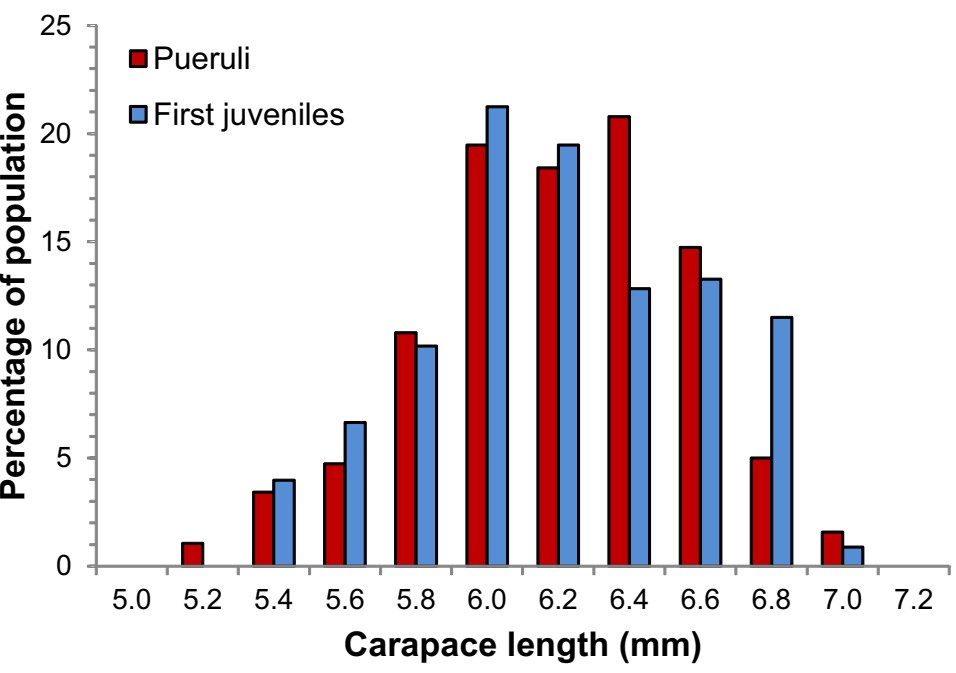

**Figure 2** **Size distribution of pueruli and first juveniles of *Panulirus argus*.** Size (carapace length) distribution of the sample of pueruli (red columns, *N* = 380) and first-stage juveniles (blue columns, *N* = 226) of *Panulirus argus* collected at Puerto Morelos and Bahía de la Ascensión, Mexico, between August 2012 and September 2013.

### Seasonal variation in CL, W, and W/CL index

Between 20 and 45 pueruli per season, and between 12 and 35 first juveniles per season were obtained from samples at Puerto Morelos between autumn 2010 and summer 2013. Results of GLMs on these data showed a significant effect of both main factors (stage and season) but not of their interaction, on CL, W and W/CL (Table 4). First juveniles had higher mean values of all three variables than pueruli, with a similar trend for both developmental stages over the sampling period (Fig. 4). For both stages, all three variables were low in summer 2011 and tended to decrease between spring 2012 and summer 2013 (Fig. 4). Results of ANCOVA showed that, for each developmental stage, the effect of season on W/CL remained significant after controlling for the significant effect of CL (the covariate) (Table 5). However, the modified W/CL index of pueruli only differed significantly from the average value in two seasons: winter 2011 (higher), and summer 2013 (lower) (Fig. 5A), whereas within-season variability (seasonal CVs) ranged from 10.1% to 11.6%. For first juveniles, the modified W/CL index was more variable over time, with higher than average values in winter and spring 2011, and lower than average values in autumn 2011 and winter 2012 (Fig. 5B), whereas within-season variability ranged from 6.7% to 7.7%.

## DISCUSSION

We compared several body dimensions and morphometric relationships between pueruli and first juveniles of *Panulirus argus*, and examined seasonal variation in a morphometric

**Table 1  Effects of developmental stage and location on body dimensions.** Results of separate general lineal models testing for effects of developmental stage (two levels, pueruli and first juveniles) and location (two levels, Puerto Morelos and Bahía de la Ascensión) on five body dimensions of individuals collected between August 2012 and September 2013: carapace length (CL), weight (W), total length (TL), carapace height (CH) and carapace width (CW).

| Body dimension | Effect | DF | Mean square | F | p |
|---|---|---|---|---|---|
| CL (mm) | Intercept | 1 | 17,695.0 | 130,033.3 | <0.001 |
| | Location | 1 | 0.414 | 3.044 | 0.082 |
| | Stage | 1 | 0.011 | 0.079 | 0.778 |
| | Location × Stage | 1 | 0.006 | 0.042 | 0.839 |
| | Residual | 602 | 0.136 | | |
| W (mg) | Intercept | 1 | 6,758,658.32 | 16,593.71 | <0.001 |
| | Location | 1 | 1,570.96 | 3.857 | 0.050 |
| | Stage | 1 | 265,286.52 | 651.326 | <0.001 |
| | Location × Stage | 1 | 50.24 | 0.123 | 0.726 |
| | Residual | 602 | 407.30 | | |
| TL (mm) | Intercept | 1 | 157,432.85 | 135,388.16 | <0.001 |
| | Location | 1 | 1.668 | 1.434 | 0.232 |
| | Stage | 1 | 34.352 | 29.542 | <0.001 |
| | Location × Stage | 1 | 5.150 | 4.429 | 0.046 |
| | Residual | 594 | 1.163 | | |
| CH (mm) | Intercept | 1 | 4,303.463 | 55,668.66 | <0.001 |
| | Location | 1 | 0.106 | 1.366 | 0.243 |
| | Stage | 1 | 223.055 | 2,885.394 | <0.001 |
| | Location × Stage | 1 | 0.461 | 5.962 | 0.015 |
| | Residual | 602 | 0.077 | | |
| CW (mm) | Intercept | 1 | 7,087.67 | 112,135.68 | <0.001 |
| | Location | 1 | 0.006 | 0.092 | 0.761 |
| | Stage | 1 | 22.28 | 352.493 | <0.001 |
| | Location × Stage | 1 | 0.02 | 0.317 | 0.574 |
| | Residual | 602 | 0.063 | | |

condition index in both stages. When comparing data from Puerto Morelos and Bahía de la Ascensión, location had no effect on any of the measured body dimensions. Stage significantly affected all dimensions with the exception of CL, which remained virtually unchanged after the post-settlement molt, as previously reported by *Lewis, Moore & Babis (1952)*. In contrast, *Abrunhosa & Kittaka (1997)* reported an increase in mean CL of 7.2% between the puerulus and first juvenile of *P. cygnus*, and *Groeneveld et al. (2010)* reported an increase in mean CL of 4–16% in *Jasus lalandii*. Although CW and TL were larger in first juveniles than in pueruli, the main differences between the two stages were an increase in CH and in W after the post-settlement molt. The increase in CH reflects the change from the relatively flattened body of the puerulus, which provides a more streamlined form

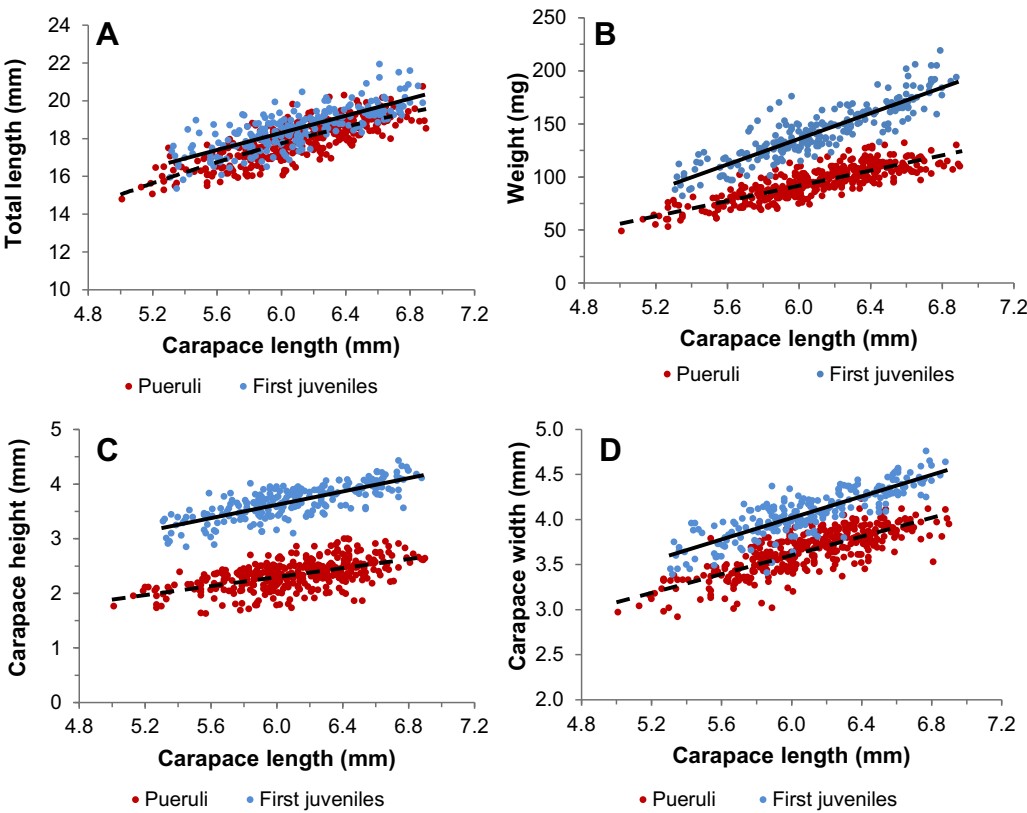

**Figure 3 Morphometric relationships of pueruli and first juveniles.** Relationships between (A) total length and carapace length (CL), (B) weight and CL, (C) carapace height and CL, and (D) carapace width and CL, with ordinary least-squares regression lines, in pueruli (red dots and dashed lines, $N = 380$) and first juveniles (blue dots and continuous lines, $N = 226$) of *Panulirus argus* collected at Puerto Morelos and Bahía de la Ascensión, Mexico, between August 2012 and September 2013.

for this forward-swimming stage, to the more cylindrical body of the benthic juveniles, which lose the ability to swim forward. This ontogenetic change was previously observed by *Grobler & Ndjaula (2001)* and *Groeneveld et al. (2010)* in *J. lalandii*. The increase in weight is greatly due to the uptake and incorporation of minerals into the cuticle of the first juveniles, which becomes thicker and heavier (*Lemmens, 1995*), and also potentially to the development of internal structures (*Lemmens & Knott, 1994*).

In the morphometric analyses, all dimensions increased linearly with CL and the slopes of all OLS regressions differed between stages with the exception of the TL vs CL regression, in which the intercepts differed significantly. Although the relationship W vs CL is typically described with a power function in spiny lobsters (e.g., *Zetina-Moguel, Ríos-Lara & Cervera-Cervera, 1996*; *Cruz et al., 2007*), we used a linear function for this relationship because of the small range in size and weight of pueruli and first juveniles. The small measured range gave rise to wide variability in most relationships (values of $r^2 < 0.7$), as was also the case for W vs CL relationships in pueruli of *J. lalandii* (*Grobler & Ndjaula, 2001*; *Groeneveld et al., 2010*). Because allometry was examined for each stage, we expected

**Table 2  Parameters of ordinary least-squares regressions between body dimensions.** Parameters (with 95% confidence intervals, CI) of ordinary least-squares regressions with untransformed values of dependent (Y) and independent (X) variables, describing relationships between measurements of body size (mm) and weight (mg) for pueruli and first juveniles of *Panulirus argus* collected in Puerto Morelos and Bahía de la Ascensión between August 2012 and September 2013. Regressions were computed separately for pueruli and first-stage juveniles and the slopes statistically compared between developmental stages; when slopes did not differ significantly, intercepts were compared.

| Y | X | Stage | Intercept $a$ (95% CI) | Slope $b$ (95% CI) | N | $r^2$ |
|---|---|---|---|---|---|---|
| TL | CL | Puerulus | 3.80 (2.70, 4.89)[***] | 2.32 (2.14, 2.50)[ns] | 372 | 0.634[***] |
| | | First juvenile | 4.71 (3.22, 6.21)[***] | 2.27 (2.02, 2.51)[ns] | 226 | 0.600[***] |
| CH | CL | Puerulus | −0.18 (−0.55, 0.19) | 0.41 (0.35, 0.47)[***] | 380 | 0.321[***] |
| | | First juvenile | −0.04 (−0.45, 0.37) | 0.61 (0.54, 0.68)[***] | 226 | 0.591[***] |
| CW | CL | Puerulus | 0.47 (0.22, 0.72) | 0.52 (0.48, 0.56)[*] | 380 | 0.620[***] |
| | | First juvenile | 0.44 (0.14, 0.74) | 0.60 (0.55, 0.65)[*] | 226 | 0.718[***] |
| W | CL | Puerulus | −123.41 (−137.99, −108.83) | 35.84 (33.46, 38.23)[***] | 380 | 0.697[***] |
| | | First juvenile | −227.22 (−253.27, −201.16) | 60.53 (56.27, 64.79)[***] | 226 | 0.778[***] |

Notes.

CL, carapace length; CH, carapace height; CW, carapace width; TL, total length; W, body weight.

[*]$p < 0.05$, [***]$p < 0.001$, [ns] = not significant.

**Table 3  Parameters of reduced major axis regressions between body dimensions and tests for allometry.** Parameters (with 95% confidence intervals, CI) of reduced major axis regressions with Ln-transformed values of dependent (Y) and independent (X) variables for pueruli and first juveniles of *Panulirus argus* collected in Puerto Morelos and Bahía de la Ascensión between August 2012 and September 2013. Regressions were computed separately for pueruli and first-stage juveniles. The last three columns refer to the type of allometry of length dimensions tested against a $H_0 : b = 1$, and of weight tested against a $H_0 : b = 3$ with Student's $t$ tests.

| Y | X | Stage | Intercept $a$ (95% CI) | Slope $b$ (95% CI) | N | $r^2$ | $t$ | $p$ | Allometry |
|---|---|---|---|---|---|---|---|---|---|
| Ln TL | Ln CL | Puerulus | 1.096 (0.991, 1.205) | 0.990 (0.930, 1.049) | 372 | 0.640 | −0.307 | 0.380 | Isometry |
| | | First juvenile | 1.175 (1.030, 1.331) | 0.965 (0.878, 1.045) | 226 | 0.597 | −0.806 | 0.211 | Isometry |
| Ln CH | Ln CL | Puerulus | −2.625 (−2.924, −2.286) | 1.920 (1.734, 2.083) | 380 | 0.315 | 9.951 | <0.001 | Positive |
| | | First juvenile | −1.114 (−1.311, −0.916) | 1.336 (1.227, 1.443) | 226 | 0.584 | 5.477 | <0.001 | Positive |
| Ln CW | Ln CL | Puerulus | −0.723 (−0.849, −0.574) | 1.117 (1.034, 1.186) | 380 | 0.625 | 3.148 | <0.001 | Positive |
| | | First juvenile | −0.515 (−0.656, −0.380) | 1.062 (0.989, 1.139) | 226 | 0.711 | 1.560 | 0.060 | Isometry |
| Ln W | Ln CL | Puerulus | −0.636 (−0.924, −0.320) | 2.867 (2.691, 3.025) | 380 | 0.716 | −1.625 | 0.052 | Isometry |
| | | First juvenile | −0.437 (−0.768, −0.105) | 2.976 (2.794, 3.158) | 226 | 0.783 | −0.252 | 0.401 | Isometry |

Notes.

CL, carapace length; CH, carapace height; CW, carapace width; TL, total length; W, body weight.

relationships between log-transformed variables to be isometric. This was the case for all relationships except for Ln CH vs Ln CL, which exhibited positive allometry in both stages, and Ln CW vs Ln CL, which exhibited positive allometry in pueruli. These results suggest that larger pueruli emerge from metamorphosis with a proportionally broader carapace than smaller pueruli, potentially reflecting individual heterogeneity among phyllosomata.

Morphometric changes between the puerulus and the first juvenile could be considered as subtle compared with the dramatic morphological change from phyllosoma to puerulus, but the post-settlement molt does involve considerable structural and morphological changes (*Lewis, Moore & Babis, 1952*; *Lemmens & Knott, 1994*; *Nishida, Takahashi & Kittaka, 1995*;

**Table 4  Effects of developmental stage and season on CL, W, and the W/CL index.** Results of separate General Lineal Models testing for the effects of developmental stage (two levels, pueruli and first juveniles) and season (12 levels, from autumn 2010 to summer 2013) on three response variables: carapace length (CL, mm), weight (W, mg), and W/CL index (condition factor) obtained from individuals collected at Puerto Morelos from autumn 2010 to summer 2013.

| Response variable | Effect | DF | MS | F | p |
|---|---|---|---|---|---|
| CL | Intercept | 1 | 20,185.082 | 191,074.299 | <0.001 |
|  | Stage | 1 | 0.604 | 5.714 | 0.017 |
|  | Season | 11 | 0.852 | 8.067 | <0.001 |
|  | Stage × Season | 11 | 0.126 | 1.191 | 0.290 |
|  | Residual | 557 | 0.106 | | |
| W | Intercept | 1 | 7,761,872 | 24,579.136 | <0.001 |
|  | Stage | 1 | 326,130 | 1,032.741 | <0.001 |
|  | Season | 11 | 3,437 | 10.884 | <0.001 |
|  | Stage × Season | 11 | 471 | 1.492 | 0.130 |
|  | Residual | 557 | 316 | | |
| W/CL | Intercept | 1 | 203,578.6 | 44,111.986 | <0.001 |
|  | Stage | 1 | 7,855.4 | 1,751.937 | <0.001 |
|  | Season | 11 | 39.9 | 10.472 | <0.001 |
|  | Stage × Season | 11 | 9.1 | 1.506 | 0.125 |
|  | Residual | 557 | 4.6 | | |

*Abrunhosa & Kittaka, 1997*; *Jeffs, Nichols & Bruce, 2001*) not necessarily evident in the body dimensions that we measured. Interestingly, *Ventura et al. (2015)* found that the genes expressed during the molt between puerulus and first juvenile in *Sagmariasus verreauxi* are the same genes expressed during metamorphosis in other crustaceans, whereas the more dramatic phyllosoma-puerulus morphological shift relies on a different, yet to be identified metamorphic mechanism.

Temporal variations in carapace length (CL) of pueruli have been previously documented in spiny lobsters (e.g., *J. edwardsii*: *Booth & Tarring, 1986*; *J. lalandii*: *Grobler & Ndjaula, 2001*; *Groeneveld et al., 2010*; *P. interruptus*: *Guzmán-Del Próo et al., 1996*; *P. argus*: *Yeung et al., 2001*). In New Zealand, pueruli of *J. edwardsii* were larger in winter than in the rest of the seasons (*Booth & Tarring, 1986*), whereas in Namibia (*Grobler & Ndjaula, 2001*) and South Africa (*Groeneveld et al., 2010*), pueruli of *J. lalandii* were larger in late summer. In all these cases, the authors offered as possible explanations that phyllosomata metamorphosing in that particular season either found more favorable feeding conditions or had more time to feed and grow, or that pueruli arriving to the settlement habitats in different seasons may have originated from different adult populations.

In the present study, the mean CL and W of pueruli of *P. argus* varied throughout the study period, with the lowest values in summer 2011 and in spring-summer 2013. In pueruli of *P. argus* collected between June 1997 and June 1999 in Florida, the smaller mean CL values also occurred during the summer (*Yeung et al., 2001*), a result that was ascribed to variable seawater temperatures and nutritional conditions during larval

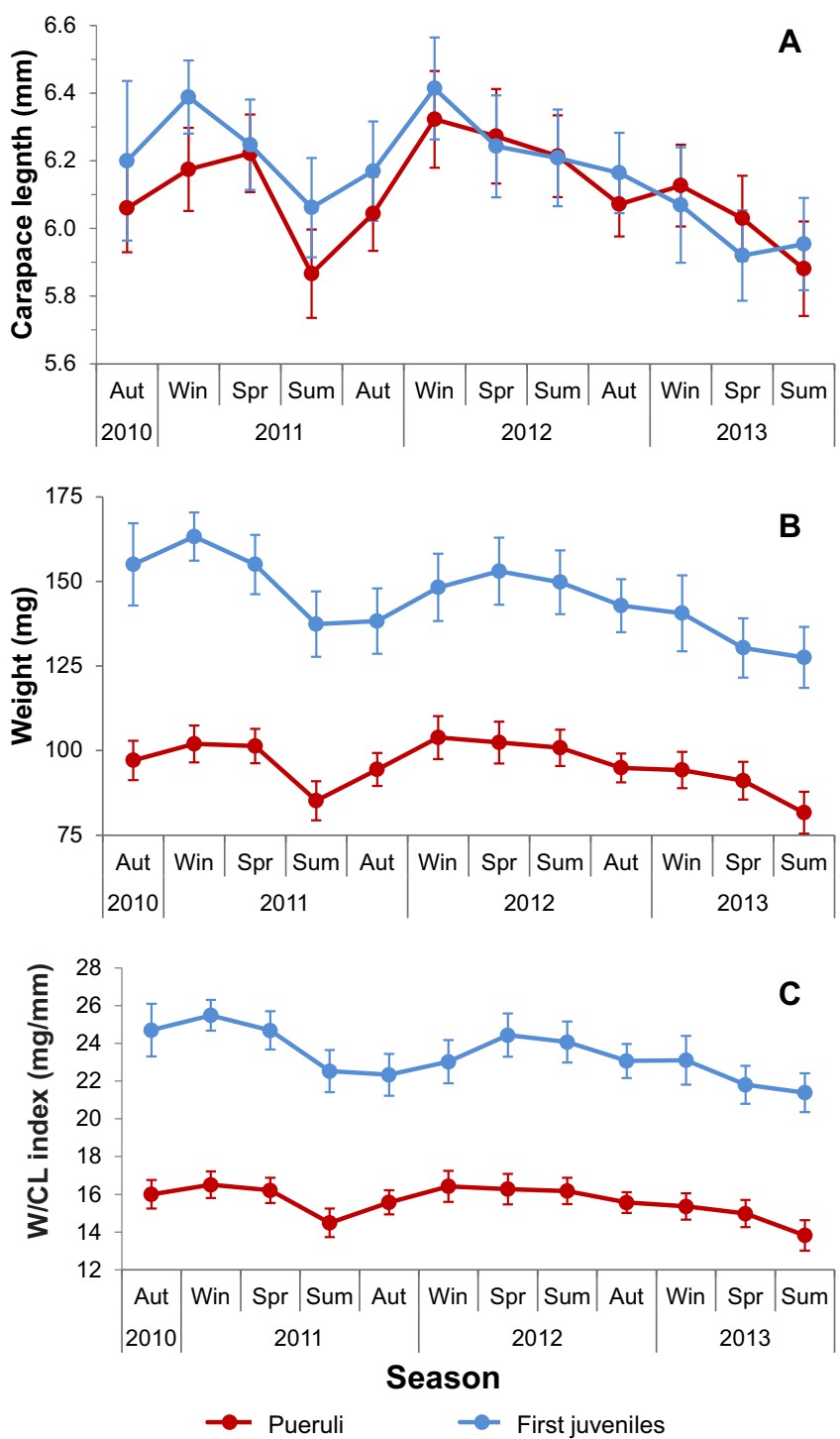

**Figure 4** **Seasonal variation in size, weight, and condition factor (W/CL index).** Seasonal variation in (A) size (carapace length, CL), (B) total weight (W) and (C) the W/CL ratio of pueruli (red dots and lines) and first juveniles (blue dots and lines) of *Panulirus argus* collected at Puerto Morelos, Mexico, between autumn 2010 and summer 2013. Error bars are 95% confidence intervals. Seasonal Ns for pueruli: 20–45; for first juveniles: 12–35.

**Table 5 Effects of carapace length and season on the W/CL index.** Analyses of covariance to test for effects of carapace length (CL, covariate) and season (categorical predictor with 12 levels: autumn 2010 to summer 2013) on the morphometric condition index W/CL (weight/CL) of (A) pueruli and (B) first juveniles of *Panulirus argus* collected from Puerto Morelos from autumn 2010 to summer 2013.

| Effect | *df* | MS | *F* | *p* |
|---|---|---|---|---|
| (A) Pueruli: | | | | |
| Intercept | 1 | 2.030 | 0.750 | 0.387 |
| Carapace length | 1 | 263.328 | 97.302 | <0.001 |
| Season | 11 | 5.690 | 2.103 | 0.020 |
| Residual | 316 | 2.706 | | |
| | | | | |
| (B) First juveniles: | | | | |
| Intercept | 1 | 88.985 | 33.277 | <0.001 |
| Carapace length | 1 | 812.965 | 304.021 | <0.001 |
| Season | 11 | 14.211 | 5.314 | <0.001 |
| Residual | 239 | 2.674 | | |

growth. *Espinosa-Magaña et al. (in press)* collected nektonic pueruli of *P. argus* offshore of the Mexican Caribbean coast in autumn 2012 and spring 2013, and found that the pueruli were larger and contained more lipid reserves in spring than in autumn. These differences were ascribed to a greater availability of basal food resources (phytoplankton) across the western Caribbean between October and March than during April to September, as reported by *Melo-González et al. (2000)*, or to the higher water temperatures during the summer, which may reduce the metabolic efficiency and the increase in size at molt of late-stage phyllosomata (e.g., *Matsuda & Yamakawa, 1997*), potentially resulting in smaller pueruli during the summer months. Indeed, in laboratory experiments, *Fitzgibbon & Battaglene (2012)* found a downward shift in the optimum temperature for late-stage phyllosomata compared to early- and mid-stage phyllosomata of *S. verrauxi*. This shift involves changes in feeding and energy metabolism, resulting in larger pueruli at lower temperatures.

*Limbourn et al. (2009)* found a significant spatial and temporal variation in total lipid content and fatty acid composition of pueruli and first juveniles of *P. cygnus* in western Australia, but to the best of our knowledge, the present study is the first examining temporal variability in a morphometric condition factor for pueruli and first juveniles of any spiny lobster species. *Robertson, Butler & Dobbs (2000)* experimentally demonstrated that the W/CL index reflected the nutritional condition of early benthic juveniles of *P. argus*, which is why we chose this index. However, as the W/CL index increases with size (*Briones-Fourzán, Baeza-Martínez & Lozano-Álvarez, 2009*), we modified this index by controlling for the significant effect of size. The modified W/CL index of pueruli was higher than average in winter 2011 and lower than average in summer 2013, yet compared with values for first juveniles, values for pueruli changed relatively less between seasons but exhibited greater within-season variability. This result would be consistent with variations in nutritional condition of late stage phyllosomata prior to metamorphosis in oceanic waters and in the distances potentially swum by individual pueruli to the settlement habitats

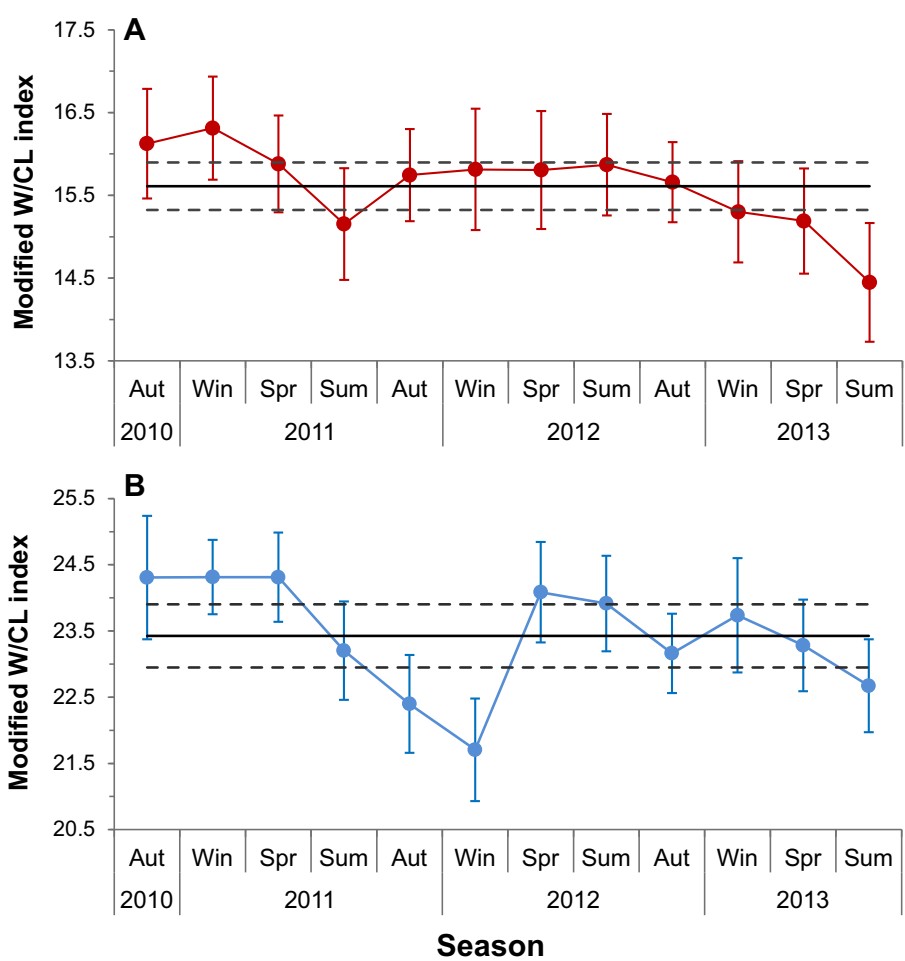

**Figure 5  Seasonal variation of modified condition factor (W/CL index).** Seasonal variation of the modified condition factor (W/CL index) after controlling for the significant effect of size (CL, covariate) in (A) pueruli and (B) first juveniles of *Panulirus argus* collected at Puerto Morelos, Mexico, between autumn 2010 and summer 2013. Error bars are 95% confidence intervals (CI). In (A–B), the overall mean modified W/CL index and its 95% CI are denoted by the black continuous and dashed lines, respectively. Seasonal Ns for pueruli: 20–45; for first juveniles: 12–35.

(*Espinosa-Magaña et al., in press*). Genetic variability could also account for some of this variation, as the northern portion of the Mexican Caribbean has been identified as a sink area potentially receiving postlarvae of *P. argus* from multiple Caribbean locations (*Briones-Fourzán, Candela & Lozano-Álvarez, 2008*; *Kough, Paris & Butler IV, 2013*). Another factor possibly involved in the variability in the W/CL index of benthic pueruli is the seasonal variation in the strength of the Yucatan current, which runs very close to the eastern coast of the Yucatan peninsula. This current is one of the swiftest western boundary currents in the world, but exhibits greater velocities in spring-summer than in autumn-winter (*Cetina et al., 2006*; *Carrillo et al., 2015*). As the nektonic pueruli would have to cross this current in order to arrive to our sampling locations, this may result in a greater expense of energy resources when the current is stronger (*Espinosa-Magaña et al., in press*).

In contrast with pueruli, the modified W/CL index of first juveniles exhibited more between- and less within-season variability. Because first juveniles need to resume feeding after the post-settlement molt to restore energy reserves, these results suggest that their nutritional condition depends to some extent on variation of local conditions of the settlement habitat, such as food availability and predation mortality. In particular, although small juveniles invest more in defensive structures, including disruptive coloration, than larger conspecifics (*Anderson et al., 2013*), the risk of predation is ever present and can inhibit foraging activity of small juveniles (*Smith & Herrnkind, 1992*; *Weiss, Lozano-Álvarez & Briones-Fourzán, 2008*) even if food is as readily available as in pueruli collectors. Yet, first juveniles of *P. argus* can starve for up to 12 days and still be able to molt if they can resume feeding after this period (*Espinosa-Magaña, Lozano-Álvarez & Briones-Fourzán, 2017*). In addition to the potential effect of local conditions, there might be a meaningful carry-over effect of the previous condition after the post-settlement molt. This is an important issue to consider because conspecific individuals can differ widely in their metabolic phenotype (i.e., their rate of energy metabolism) (*Burton et al., 2011*), and this phenotypic variation can be more important than variation in larval supply in regulating marine populations (*Marshall & Morgan, 2011*).

## CONCLUSIONS

Pueruli and first juveniles of *P. argus* had a similar range in CL, but the latter were heavier and had a greater CH than the former. As predicted, there was significant seasonal variation in CL, W, and the W/CL index (a morphometric condition factor) of both stages. However, within-season variation in the modified W/CL index was greater for pueruli than for first juveniles, potentially reflecting variability in the condition of larvae prior to metamorphosis and in the distances swum by individual pueruli to the settlement habitat, whereas between-season variation was greater for first juveniles than for pueruli, suggesting a combination of carry-over effects and variation in the effects of local factors (e.g., predation risk and food availability). Whether such differences have any impact on survival of these stages of *P. argus* remains to be determined. For example, validating the results of W/CL index of pueruli and first juveniles with biochemical analyses (lipid and protein contents) would support their continued use to monitor nutritional condition of these stages over the long term, whereas experimental studies could help determine how the condition factor of first juveniles varies in the presence/absence of predation risk at different levels of food availability. These and other investigations are necessary to unravel the complexity of linkages between factors potentially decoupling settlement from recruitment processes.

## ACKNOWLEDGEMENTS

We gratefully acknowledge the invaluable technical assistance provided by F Negrete-Soto and C Barradas-Ortiz throughout the study. Additional assistance in field and/or laboratory work was provided by R Muñoz de Cote-Hernández, A Espinosa-Magaña,

R Candia-Zulbarán, R González-Gómez, L Cid-González, N Luviano-Aparicio, JP Huchin Mian, and IH Segura García.

### Funding

This work was funded by Consejo Nacional de Ciencia y Tecnología (CONACYT, México) (project CB-101200, granted to Patricia Briones-Fourzán) and Universidad Nacional Autónoma de México. Rogelio Martínez-Calderón received an MSc scholarship from CONACYT. The funders had no role in study design, data collection and analysis, decision to publish, or preparation of the manuscript.

### Grant Disclosures

The following grant information was disclosed by the authors:
Consejo Nacional de Ciencia y Tecnología (CONACYT, México): CB-101200.
Universidad Nacional Autónoma de México.

### Competing Interests

The authors declare there are no competing interests.

### Author Contributions

- Rogelio Martínez-Calderón conceived and designed the experiments, performed the experiments, analyzed the data, prepared figures and/or tables, authored or reviewed drafts of the paper, approved the final draft.
- Enrique Lozano-Álvarez conceived and designed the experiments, analyzed the data, contributed reagents/materials/analysis tools, authored or reviewed drafts of the paper, approved the final draft.
- Patricia Briones-Fourzán conceived and designed the experiments, analyzed the data, contributed reagents/materials/analysis tools, prepared figures and/or tables, authored or reviewed drafts of the paper, approved the final draft.

### Field Study Permissions

The following information was supplied relating to field study approvals (i.e., approving body and any reference numbers):
A permit for collection was issued by Comisión Nacional de Acuacultura y Pesca (DGOPA-06695.190612.1737).

### Data Availability

The raw morphometric data are provided in Data S1.

### Supplemental Information

Supplemental information for this article can be found online at http://dx.doi.org/10.7717/peerj.5297#supplemental-information.

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
