# Peer review of "Morphometric relationships and seasonal variation in size, weight, and a condition index of post-settlement stages of the Caribbean spiny lobster"

_PeerJ, doi:10.7717/peerj.5297_

## Round 0.1 · original submission · Major Revisions

Though generally positive about your work, the Reviewers have raised a number of concerns about the manuscript that need to be addressed. I have gone over the manuscript myself in some detail and I’ve summarized what I view as the most critical concerns below.

I invite you to submit a revision addressing these concerns. You should detail the changes you have made and specifically explain how you have addressed the reviewer’s points. Since the revisions may require some degree of re-analysis of the data, I consider this a major revision.

1) The reviewers all commented favorably on the literature review, but Reviewer #3 identified Anderson et al. 2013 as an especially relevant article that was not cited. I believe this is: Anderson, J.R., Spadaro, A.J., Baeza, J.A., & D.C. Behringer. 2013. Ontogenetic resource allocation shifts in defensive structures of the Caribbean spiny lobster, Panulirus argus (Latreille, 1804). Biological Journal of the Linnean Society 108: 87-98. Although the work refers mainly to larger stages, you will see the relevance. Please consider incorporating it into your review.

2) There are a number of methodological points to be clarified and discussed:
a. Two of the three reviewers were confused by the sampling. Although there is a comment in the Methods (l. 163-165), it is obviously not clear enough. It may help to separate the work to two separate studies: a more preliminary one that aims to identify cohorts and evaluated two sites, and a second conducted at a single site that has a seasonal component. Clarifying which data from which study are presented in which figures might be helpful.

b. Reviewer #2 would like to see comments regarding the accuracy/reproducibility of the size and mass measurements.

c. Two of the three reviewers wanted clarifications regarding the separation of larval stages and the terms used to refer to them. Reviewer #2 points out that grouping pigmented and unpigmented pueruli as “benthic” is problematic. Since it’s impossible to place them in the water column, I concur. What’s wrong with simply calling them “pueruli”? Reviewer #1 asks for clarification about how stage 1 and stage 2 (older?) juveniles were distinguished (lines 154-155) and makes the intriguing suggestion that a second mode in Figure 2 data suggests the separation was not achieved. This should be explained and the possibility discussed.

d. Reviewer #3 is concerned that since collectors were harvested monthly, some of the individuals may have effectively ‘starved’ for up to 30 d, altering their condition substantially. This should be addressed in the discussion.

e. A clear explanation/rationalization for the division of the year into “seasons” by the calendar should be provided (see comments of Reviewer #1). This is especially important because in the Caribbean, months like September (presumably a “autumn” month by the calendar?) are typically much more similar to preceding months than following ones in terms of oceanographic conditions; would astronomical distinctions (e.g. September 21 as the division between summer and autumn) produce different results? In any case, clearly labelling the month samples on Figure 3 and 4 would allow a reader to make up their own minds.

f. Allometric regression analyses (lines 195-6) follow Hartnoll (1982), but there is a potential issue. In such regressions, both X and Y variable have error associated with them. This violates the assumptions of a normal (Type I) regression and requires a different approach: a Type II (Reduced Major Axis) regression, described by Laws & Archie (1981) (Laws, E.A. & J.W. Archie. 1981. Appropriate use of regression analysis in marine biology. Mar. Biol. 65: 13-16.). There might be arguments against doing this, but if so, the authors should make them in the manuscript.

3) Overall, the manuscript doesn’t come to clear conclusions. As Reviewer #3 points out, the current conclusions simply summarize the results again. The authors state a purpose and a prediction in the Introduction (lines 130-140). They should instead return to these in the conclusion. Reviewer #3 also suggest posing (and answering) the question: “which morphometric parameters should be associated with enhanced or diminished lobster recruitment?” If the question can’t be answered because of ‘decoupling’, then the authors have the opportunity to make specific recommendations of what the next step should be. This will allow the authors to share their expertise and greatly improve the value of the manuscript.

4) Line 186. “covariate” not “covariant”
Figure 2 legend. Italicize Panulirus argus
Figure 2. Y-axis legend “percentage of population” versus “Percent lobsters”
Figure 4 legend. “corrected W/CL” I am not sure that this is the best way define or present these data. They represents the results of an Analysis of Covariance, which attempts to remove the global effect of CL on W and thus what remains is a clearer indicator or condition. It may be useful to see how other publications have done this. The units of “mg/mm” can certainly not be correct.

·

Basic reporting

Clear unambiguous professional English used throughout:
- Well-written and easy to read, with some caveats.
Minor inconsistencies, such as using (or not) abbreviations for weight (W), carapace height (CH) etc. See for example between lines 236 and 249.
In several areas, the writing could be more concise, for example:
Line 150: Delete ‘For the present study’ and start the sentence with “We used samples…
Lines 209-210: Full-stop after (Figure 2). Then delete …’shows the great overlap in size range between stages.’ It is quite obvious from the preceding ranges and the Figure.
Line 285: replace the winding sentence with: ‘The small measured range gave rise to wide variability in most relationships (r2 <0.5) ….’
- Intro and background show context:
The Intro is well-written and structured, and provides the required context.
Line 82: Delete ‘fighting ability’ which does not make sense here. Maybe in other taxa, or life stages, but probably not in pueruli or 1st stage juveniles.
The references cover the field well. Throughout the paper, the standard of one author followed by et al. should be used (not 2 authors + et al.).
Line 136: What would ‘local effects after the post-settlement molt’ be? A sentence to be added to explain likely local effects.

The structure does not always conform to the norm, with parts of the Materials and methods appearing in the Results section and vice versa.
Lines 171-174: Move to Results section.
Lines 174-176: Delete. It is already in the results section (see lines 210-212).
Lines 178-181: Delete. Already stated in lines 158-159. Elaborate the initial (158) text to define the calendar seasons – which months in each season and why.
Lines 189-191: Delete. Already stated in lines 163-165
Line 194-195: Delete the sentence…’As the effect of location was not significant…’ This is already in the Results section (lines 230-235), where it fits much better.
Line 214-216: This needs to be moved to the material and methods to describe the seasons that were selected (see comment above)
The Figures and Tables are relevant and good quality. The raw data has been supplied in an excel file.

Experimental design

The research questions are well defined, and relevant. The manuscript addresses an important knowledge gap in a novel way.
The experimental design is generally rigorous and performed to a high standard. There remain some points to be considered in more depth by the authors:
- Selection of seasons: In tropical ecosystems, seasonal effects can become blurred, and in many cases studies rely on dominant environmental drivers to define ecologically relevant seasons, for example monsoon seasons; or wet / dry periods etc (see Munga et al. (2013) – Species composition, distribution patterns and population structure of penaeid shrimps in Malindi-Ungwana Bay, Kenya, based on experimental bottom trawl surveys. Fisheries Research 147:93-102). The authors should define their calendar seasons (which months incorporated in each season) and also motivate exactly why they chose these specific intervals. Alternatively, investigate whether there are environmental drivers that would better describe seasonal effects on larval settlement, rather than simply splitting the year into four 3-month groups.
- Distinguishing between 1st and 2nd stage juveniles: There will certainly be overlap in the size-ranges of 1st and 2nd stage juveniles in nature. How did the authors decide which specimens were in the 1st stage (and thus included in the study) or 2nd stage (and excluded)? This needs to be explained in the Materials and Methods. The question is pertinent, because in Figure 2, there appears to be 2 modes in the category for 1st juveniles. Note that there are single modes (and roughly normal size distributions) for transparent- and pigmented pueruli, respectively. The second mode might reflect the inclusion of 2nd stage juveniles.
The methods have been described with sufficient detail –but with some redundancy (repetitions) as indicated in 'Basic reporting' above.

Validity of the findings

Data of lobster settlement patterns are quite scarce globally, because it is labor intensive to collect and numbers that are caught on collectors are often very low (as also experienced in the present study). I commend the authors for undertaking this taxing (but important) project, and for analyzing the data in a logical and scientifically rigorous way. I have no doubt that the findings are valid, and found the conclusions to be strongly supported by the data and analyses. As with any manuscript, some improvements can be made (see comments above) – the most important ones are improving the structure of the manuscript by removing repetitions; revisiting the seasonal stratification strategy used; and checking whether inclusion of stage 2 juveniles might erroneously have contaminated the data.

Additional comments

Good work. This study fills in an important knowledge gap in a clear and unambiguous way.

Reviewer 2 ·

Basic reporting

This article included enough introduction although probably is so extended to arrive to explain the main goal of the study. Maybe exists to many supposition and idea linked to use the variable weight /Carapace length (index condition) to be used in order to understand a key factor to explain the objective that the author would like to explain.
There are too many steps to understand which are the factor why Panulirus argus is an species that have a decouple the relationship between levels of settlement and abundance of subsequent benthic are phases are to explain if the nutritional condition is one of the causes that produce the decoupling.
The index of condition used by the authors is supported by different authors and it should be considered correct, although the author don’t use any direct parameter as ( percentage of fat, tax of ingested food, proteins, or quality of food evaluated by isotopes). In the introduction the author should explain that the nutritional condition is only one of the factors to produce the die of the first juveniles, not the only direct evidence to decline the abundance of settled individuals.
All the steps to arrive at the direct results are well explained and justified by papers of the other authors published before. Although, after a complex introduction the final goal proposed at the MS (lines 75-78), seem that is not the final aim of the study. I suggested that the introduction may be more clear and don’t suppose to much direct relation effect because finally the results showed, are not directly interpreted as the objective of the paper. The results of the paper are clear and correct developed but maybe it will be considered an important new information using a non invasive technique to evaluate the nutritional condition. In any case a direct relation that explain the lack relationship between the settlement and abundance recruits.

Experimental design

I propose to add a table to explain the samples used for each analysis with a information of the location and season that are captured. It offers clear information to the set of data that is used in any cases studied.

Line 207 must to say pigmented puerulus instead of benthic pueruli

The categories of sizes classes proposed in Line 207-208 must to be explained more clear the main reason , I understand that it is due to the color of each stage, but may be need to be according to the ecological role that it plays in the settlement.

The experimental design is so confused due to the aggregation of samples used by different purposes. Line 148 say that there are two locations. Line 150 said that the samples were obtained in 1 location , in the line 151 explain that exist an additional location to get samples. Avoid the line 150., only make confusion.

Line 161 The measures used in small size individuals and for using in a index may be getting more accuracy . (0.01 mm), the same for the total weight. In a present day to use a more precision instrumentation is to easy.

Statistical analysis proposed is correct to evaluate the objectives of any specific objective findings

Validity of the findings

The findings of the paper are well described an supported by statistical analysis, although a more detail analysis if the differences between the months were the differences are find may be to analyses statistically, in that sense the result and conclusion of the role that nutritional condition of the spiny lobster will be more robust.
The findings of the morphometric relationships propably are out of the main goal of the paper and don’t contribute any essential results to the general conclusions. It gives interesting information but low information to the main objective to the paper and is poorly debated.
Figure 5 plotB probably give more information if the blue dots out of the pool of data will be marked as a different color that explain individuals of a season that weight is anormal p.e. winter 2012.

Additional comments

The MS need to be revised heavily in order to put more clearly the information used and the identification of the main objectives of the MS.
Tables and plot are generally well proposed.

Reviewer 3 ·

Basic reporting

The manuscript has quite a few grammatical issues that need cleaned up, including problems with spelling, verb tense, and sentence structure which is not unexpected if one's first language is not English. Still, the manuscript would benefit from having a native English-speaker with strong writing skills help them clean up the grammatical problems in the manuscript.

Introduction was choppy in terms of its treatment of different subjects; moved among topics and then reversed back to dwell again on an earlier topic.

Most literature seems appropriate but did not reference the Anderson et al 2013 paper on allometric growth in spiny lobsters that discusses morphometric and color changes in postlarvae, early benthic stage juveniles, and later-stage juvenile lobsters.

Conclusions were not conclusions, but instead were just a recapitulation of results.

Experimental design

Statistical analyses are severely flawed. Could have used site and year as random factors in the analysis to remove those effects from the error term, and month within season as the source of replication for season.


In the description of the study where the authors talk about seasonal effects they also mention site effects, but from their description it looked like they only used data from one site. What is going on?

lines 144-146: samples taken once a month from collectors means that animals could have been on the collectors from 1- 30d. That could affect some of these morphometric relationships, particularly the weight measure based on lipid catabolism of postlarvae and feeding by juveniles. This is a big confounding factor.

Validity of the findings

Morphometrics not linked to nutritional condition in this paper. There remains a hypothetical link between the index and nutritional condition - if anything to make this more impressive the authors should have used a biochemical analyses in concert with this morphometric analysis.

Line 104 – States that biochemical analysis can be “expensive and time-consuming” (and I agree) but would not sampling larvae also be time consuming? Perhaps not to the same extent but this sounds misleading.

Line 261 – Seems obvious that the pueruli would be morphometrically different between stages.

Line 306 – Ascribes differences to greater availability of basal food but notes above that there were no differences among locations. Would not food resources be different for different locations?

Additional comments

This manuscript touches on an interesting, but fairly well-known hypothesis (particularly in fishes) that the link between levels of settlement and abundance of benthic phases may be decoupled if there is considerable variability in nutritional condition of settling stages that translates into lower post-settlement survival. In this paper, they attempt to equate basic measures of morphology of postlarval and juvenile spiny lobsters with such a phenomenon. But the manuscript only delves into the first steps of determining whether nutrition is a plausible link, because they never link morphology, to nutrition, to survival. Thus, despite a hopeful introduction that outlines an interesting question, the study falls far short of delivering data that can be brought to bear on the question. The data simply amount to a morphometric comparision among months of postlarvae, but whether such differences correlate with nutritional condition or have any impact on survival remains unknown. Moreover, the authors do not offer any predictions as to which morphometric parameters in particular should be associated with enhances or diminished lobster recruitment.

---

## Round 0.2 · accepted · Accept

The revision address the issues raised by the reviewers. Points that were unclear have been clarified and the manuscript has been strengthened. Nice work!

# ·

Basic reporting

Basic reporting and structure have been improved. It is at an acceptable level for publication in a scientific journal. I have made minor edits to the text (MS-Word file, attached) to fix grammatical errors, or remove redundant text.

Experimental design

Good. See previous review

Validity of the findings

No comment

Additional comments

Reviewer comments on the first draft have been addressed in a satisfactory manner.